# Plasma Exchange Reduces Aβ Levels in Plasma and Decreases Amyloid Plaques in the Brain in a Mouse Model of Alzheimer’s Disease

**DOI:** 10.3390/ijms242317087

**Published:** 2023-12-04

**Authors:** Santiago Ramirez, Suelyn Koerich, Natalia Astudillo, Nicole De Gregorio, Rabab Al-Lahham, Tyler Allison, Natalia Pessoa Rocha, Fei Wang, Claudio Soto

**Affiliations:** Mitchell Center for Alzheimer’s Disease and Related Brain Disorders, Department of Neurology, McGovern Medical School, University of Texas Health Science Center, Houston, TX 77030, USA; santiago.d.ramirez@uth.tmc.edu (S.R.); suelyn.koerich@uth.tmc.edu (S.K.); natalia.v.astudillocorral@uth.tmc.edu (N.A.); nicole.f.degregorio@uth.tmc.edu (N.D.G.); rabab.allaham@uth.tmc.edu (R.A.-L.); tyler.allison@uth.tmc.edu (T.A.); natalia.pessoarocha@uth.tmc.edu (N.P.R.); fei.wang.1@uth.tmc.edu (F.W.)

**Keywords:** Alzheimer’s disease, plasma exchange, albumin, amyloid-β, amyloid plaque, plasmapheresis

## Abstract

Alzheimer’s disease (AD) is the most common type of dementia, characterized by the abnormal accumulation of protein aggregates in the brain, known as neurofibrillary tangles and amyloid-β (Aβ) plaques. It is believed that an imbalance between cerebral and peripheral pools of Aβ may play a relevant role in the deposition of Aβ aggregates. Therefore, in this study, we aimed to evaluate the effect of the removal of Aβ from blood plasma on the accumulation of amyloid plaques in the brain. We performed monthly plasma exchange with a 5% mouse albumin solution in the APP/PS1 mouse model from 3 to 7 months old. At the endpoint, total Aβ levels were measured in the plasma, and soluble and insoluble brain fractions were analyzed using ELISA. Brains were also analyzed histologically for amyloid plaque burden, plaque size distributions, and gliosis. Our results showed a reduction in the levels of Aβ in the plasma and insoluble brain fractions. Interestingly, histological analysis showed a reduction in thioflavin-S (ThS) and amyloid immunoreactivity in the cortex and hippocampus, accompanied by a change in the size distribution of amyloid plaques, and a reduction in Iba1-positive cells. Our results provide preclinical evidence supporting the relevance of targeting Aβ in the periphery and reinforcing the potential use of plasma exchange as an alternative non-pharmacological strategy for slowing down AD pathogenesis.

## 1. Introduction

Alzheimer’s disease (AD) was described in 1907 as an “unusual illness of the cerebral cortex” [1]. AD is a fatal neurodegenerative disorder and the leading cause of dementia, affecting more than 45 million people worldwide [2,3,4]. AD prevalence is projected to reach 115 million people in 2050, but this number may rise to 152 million [3] due to the projected increase in life expectancy. This situation is not different in the United States; the Alzheimer’s Association estimated an AD prevalence of 6.7 million individuals among Americans 65 years of age and older in the U.S. in 2023, leading to an estimated USD 345 billion in healthcare expenses. AD prevalence could rise to 13.8 million by 2060, with a subsequent increase in the burden on the health care system [4]. AD is pathologically characterized by the presence of abnormal protein deposits consisting of neurofibrillary tangles composed of hyperphosphorylated Tau protein [5,6] and amyloid plaques composed mainly of the amyloid-β peptide (Aβ) [7,8,9]. Although the exact role of amyloid accumulation in AD pathogenesis is not completely understood, compelling evidence suggests that the accumulation of Aβ aggregates may initiate the AD cascade, preceding tau deposition and subsequent brain damage [10]. Accordingly, reducing Aβ levels in the brain has been widely considered an attractive therapeutic strategy for AD, one mainly carried out through the use of anti-Aβ monoclonal antibodies [11,12,13,14].

One of the mechanisms of Aβ clearance from the brain is transport across the blood vessel walls into the circulation [15]. It has been shown that Aβ can cross the blood–brain barrier in both directions [15], leading to the hypotheses positing that there is an equilibrium between the brain and blood pools of Aβ and that removing Aβ from plasma might extract Aβ from the brain [16,17]. Moreover, it has been shown that brain-derived Aβ can be cleared via the liver, kidneys, gastrointestinal tract, and skin [18,19,20]. In this context, we have previously demonstrated that Aβ deposition in the brain can be induced from different peripheral routes [21], including the intravenous administration of Aβ aggregates [22]. We have also shown that performing whole-blood exchanges reduced amyloid deposits in the brain and improved the cognitive performance of a mouse model of AD (Tg2576) [23]. These findings confirmed and extended previous results showing that reducing peripheral Aβ may represent a viable strategy for AD therapy [24,25,26,27]. Here, we hypothesized that removing plasma via a monthly plasma exchange (PE) procedure would effectively decrease the progressive accumulation of Aβ in the brain. To test our hypothesis, we subjected male and female APP/PS1 mice to a series of PE procedures using a 5% albumin solution while they were 3 to 7 months of age. Our results show that PE significantly delayed the progressive accumulation of Aβ in the plasma, similar to what has been previously reported [24,25,26,27]. Strikingly, we observed a significant reduction in the insoluble Aβ fraction in the brain and a 45–50% reduction in amyloid plaque load in the mice treated with PE. Our results provide evidence of the potential benefit of PE in reducing Aβ levels in the brain via targeting the periphery.

## 2. Results

### 2.1. Plasma Exchange Prevents the Increase in Aβ Levels in Plasma

To investigate whether PE could prevent a progressive increase in Aβ levels in the plasma, we performed four PE procedures once a month starting when the mice were 3 months old (3 mo) (Figure 1). The levels of Aβ were analyzed via ELISA. We observed that plasma Aβ levels significantly increased over time in the 3 mo control vs. the 7 mo control; (390.6 ± 92.24 vs. 1107 ± 57.92 pg/mL; *p* = 0.0001). The PE procedure effectively spurred the increase in Aβ levels in plasma in mice at 7 mo. (498.5 ± 151.8 vs. 1107 ± 57.92 pg/mL; *p* = 0.0006) (Figure 2). The levels of Aβ at intermediate time points in the PE mice were comparable to the baseline levels (Appendix A).

### 2.2. Plasma Exchange Prevents the Formation of Amyloid Plaques in the Brain

After observing that PE prevents the progressive increase in Aβ levels in plasma, we conducted further evaluations to assess the effect of PE on the progression of amyloid pathology in the brains of both treated and control APP/PS1 mice. At 7 months, all animals were humanely euthanized, and their brains were processed for histology. We analyzed the characteristics of amyloid pathology using ThS amyloid staining and 4G8 immunostaining in the cerebral cortex (Figure 3a) and hippocampus (Figure 4a). Appendix A provide low-magnification scans of the whole brain from representative control and PE-treated animals, respectively. Visual inspection revealed that the PE treatment appeared to reduce the number and burden of plaques in the brain. To validate this conclusion, the cortex and hippocampus were analyzed for amyloid burden, plaque number, and plaque size. We found that the animals subjected to PE had reduced ThS burden in the cortex (47.18 ± 4.06 percent of the control; *p* = 0.0011) (Figure 3b) and the hippocampus (32.75 ± 7.071 percent of the control; *p* = 0.0016) (Figure 4b) as well as a significantly lower number of plaques in the cortex (53.22 ± 3.134 percent of the control; *p* = 0.0025) (Figure 3c) and the hippocampus (51.84 ± 7.854 percent of the control; *p* = 0.0421) (Figure 4c). The reduction in amyloid pathology measured via histological staining was supported by studies on the amount of insoluble Aβ conducted using ELISA. Our results showed that PE treatment was effective in reducing the accumulation of insoluble Aβ in the brain (386,025 ± 35,946 vs. 437,331 ± 6951 pg/mL; *p* = 0.039) (Appendix A). No differences were observed in the soluble Aβ fraction (134.7 ± 10.09 vs. 146.1 ± 8.712 pg/mL; *p* = 0.234) (Appendix A).

We further measured the plaque diameter to compare the populations of plaques below or above 50 μm. We found that PE reduced plaques with diameters both <50 μm (56.27 ± 3.447 vs. 100 ± 9.829%; *p* = 0.0094) (Figure 3d) and >50 μm in the cortex (45.44 ± 5.006 vs. 100 ± 14.19%; *p* = 0.0207) (Figure 3e). However, in the hippocampus, this effect was observed only in plaques with diameters < 50 μm (47.16 ± 6.925 vs. 100 ± 15.25%; *p* = 0.0428) (Figure 4d). No significant differences were observed in plaques with diameters >50 μm (86.79 ± 24.85 vs. 100 ± 16.41%; *p* = 0.657) (Figure 4e). Finally, we found that the burden area that was positive for the microglia marker Iba1 was significantly lower in the PE-treated animals compared to the controls (37.04 ± 17.5% vs. 89.18 ± 10.6%; *p* = 0.036) (Appendix A). A similar trend was observed for the astrocyte marker GFAP (40.4 ± 9.6% vs. 100.2 ± 25.78%; *p* = 0.069) (Appendix A).

### 2.3. Plasma Exchange Modifies the Relative Frequencies of Aβ Plaque Size

To further investigate if PE affects the formation of plaques, we performed a relative frequency distribution analysis of the plaque size in the cortex and hippocampus. Individual plaque size distribution data were subjected to a logarithmic transformation to further analyze the relative frequency distributions and compare them using a non-linear best-fit analysis. Considering that the absolute number of plaques was substantially reduced by PE, the relative frequencies of the different plaque sizes in the cortex were not affected, and a single Gaussian distribution model fitted both the PE and control distributions: Y = 0.1349 × exp(−0.5 × ((X − 1.357)/0.3017)^2^); r2 = 0.9785; *p* = 0.8113 (Figure 5a). However, in the hippocampus, PE significantly modified the size–frequency distribution, mostly via reducing the frequency of small-sized plaque populations but also via increasing the proportion of plaques with larger sizes. Different Gaussian distributions between the PE vs. control groups were observed: Y = 0.1342 × exp(−0.5 × ((X − 1.432)/0.3105)^2^); r2 = 0.895 vs. Y = 0.1319 × exp(−0.5 × ((X − 1.292)/0.3074)^2^); r2 = 0.924; *p* = 0.0034 (Figure 5b).

## 3. Discussion

In our previous work [23], we reported a proof-of-concept study on the preventive and therapeutic potential of the removal of Aβ peptide from the periphery via whole-blood exchange. We observed a reduction in progressive Aβ accumulation in the brain and improvements in cognition in an AD mouse model (Tg2576). Nevertheless, massive monthly whole-blood replacement is not feasible in medical practice; therefore, it is medically relevant to develop and evaluate a more applicable strategy. We rationalized that using PE, a procedure routinely employed in the clinical setting (known therein as plasmapheresis) [28], might be an ideal candidate for testing our concept with more clinical applicability, although not without risks [29]. We adapted our PE protocol to perform a monthly replacement of ~15% of the total plasma volume with a 5% albumin solution in a progressive mouse model of AD.

Plasmapheresis, commonly used for therapeutic PE, was developed in 1914 [30] and is primarily employed for plasma donation without requiring a replacement fluid [31]. Traditionally, it has been used to remove negative components in blood plasma and exchange them with donated blood products, with applications predominantly lying in the treatment of neurological, immunological, and hematological disorders [32]. The efficacy of PE relies on multiple factors, including the volume of removed plasma, the distribution and rate of production of pathological substances by the body, and the distribution of compounds between the plasma and other organs [33,34].

Considering the evidence suggesting that peripheral Aβ may play a role in brain pathology [16] and that deficits in Aβ clearance in the periphery may contribute to the development of AD [18,19,20], we explored the use of PE to remove Aβ in the blood, expecting to ameliorate amyloid deposition in the brain. To this end, based on previous publications showing that small-sized plaques grow faster than large-sized plaques in 6-month-old compared to 10-month-old APP/PS1 mice [35], we started the treatment at an early time point of the APP/PS1 model. We hypothesize that replacing small volumes of plasma may be more effective in allowing us to visualize the effect of the treatment in new plaque formation and plaque growth if started at the early phase of pathogenesis.

Our findings show that PE was able to maintain low Aβ levels in the plasma of APP/PS1 mice at 7 mo., maintaining levels similar to those observed at 3 mo., avoiding the expected time-dependent increase reported for this model [36] (Appendix A). Based on the levels of Aβ in plasma at 7 mo. and the expected blood and plasma volumes for a mouse [37], we estimate that the effect of PE treatment was equivalent to a reduction of 0.6 ng of Aβ, i.e., 2.5 times more than what was reported for a single peritoneal dialysis in the APP/PS1 model [26]. This result supports the biological relevance of our method in removing Aβ from the circulation. Furthermore, the brains of the mice treated with PE showed lower levels of insoluble Aβ; fewer Aβ-plaques; changes in plaque size frequency distributions; and reduced burden of activated microglia in the hippocampus. Interestingly, based on a relative frequency analysis of the plaque size, we observed that the treatment affected the hippocampus and cortex differently, reducing mostly the small-sized plaque population in the hippocampus compared to the cortex. We interpreted these results as a possible modification of the plaque growth dynamics due to PE. However, this finding could be partially explained by the different spatial patterns of amyloid deposition reported for the APP/PS1 model [38]. Nonetheless, these results suggest that PE applied at early stages in the APP/PS1 model may affect the formation of new plaques and that the lower levels of Aβ burden observed in the brain might be due, in part, to the removal of Aβ from the circulation via promoting the mobilization of Aβ peptides out of the brain and/or via relieving the clearance of Aβ in the periphery. We cannot exclude the possibility that PE may have an effect on reducing plaque formation in the brain via removing other molecules present in the plasma that could have pathophysiological roles in plaque formation. A recent report demonstrated that a deficiency in C3, a key molecule in the complement system, and various plasma proteins that induce inflammatory responses could reduce Aβ deposition in APP/PS1 mice [39]. This reduction may be attributed to improved interactions between glial cells and plaques [40]. However, it is important to note that this hypothesis has not yet been studied in the context of PE and AD.

It is important to consider the existence of other peripheral sources of Aβ, such as skeletal muscles, skin fibroblasts, and blood cells [41,42], as well as the binding of Aβ to various lipoproteins and albumin in plasma [43,44,45]. In this sense, it is relevant to mention that the use of albumin for PE, which is known to act as an Aβ transporter, a scavenger of endogenous and exogenous toxins, a modulator of the immune and inflammatory responses, and a possessor of antioxidant properties [45,46,47], could have played an active role in the beneficial effect of PE in our study. Interestingly, changes in albumin levels have been independently linked to both aging and neurodegeneration, wherein situations of low serum albumin levels can be associated with an increased risk of cognitive impairment among the elderly population, possibly via increasing Aβ deposition [48]. In addition, low albumin levels can also be associated with type 2 diabetes, a relevant risk factor for AD [49,50,51]. Therefore, albumin replacement has been already proposed as a promising therapeutic strategy for treating AD [26,34,52,53].

Besides the possibility of Aβ removal from plasma and a putative role of albumin in PE, there are other possible effects reported for this treatment that could have actively but indirectly contributed to the observed results. For example, it has been suggested that PE has the ability to reduce pro-aging phenotypes through its dilutive effect on old plasma, leading to a switch toward a younger proteome signature and a pro-regenerative phenotype [54]. In support of this, a study conducted using mouse models found that the replacement of old plasma with a 5% albumin solution induced broad rejuvenation changes in different tissues [55]. These findings suggest that PE may have additional beneficial effects beyond solely the removal of Aβ peptides.

In summary, our current findings obtained in an animal model of AD suggest that PE is a viable non-pharmacological treatment for AD that should be explored clinically. Indeed, a recent clinical trial incorporating 347 mild-to-moderate AD patients showed that PE with albumin replacement can slow the decline or stabilize AD symptoms [25]; however, the mechanism behind this is not clearly understood.

## 4. Materials and Methods

### 4.1. Animal Experiments

Hemizygous male and female APP/PS1 mice (MMRRC stock #34832) that were 3 months of age were equally and randomly distributed among the experimental groups. We used *n* = 5–10 animals per group, as indicated in each figure legend. These mice express human APP carrying the Swedish mutation and the human presenilin-1 protein carrying the DeltaE9 mutation, both related to early-onset AD. Consequently, these animals exhibit sporadic Aβ aggregate formation as early as 4–5 months of age, with a substantial accumulation and formation of amyloid plaques at ages older than 6 months [36]. Animal housing was carried out in ventilated racks at 22 °C and under a 12 h light–dark cycle. Fresh water and standard chow were available ad libitum.

### 4.2. Plasma Exchange Procedure

A surgical procedure was required in order to expose the jugular vein to grant access to the vascular compartment. Mouse anesthesia was induced with 5% isoflurane. A sterile ocular lubricant (CAT# NDC 17033-211-38, Dechra, Northwich, UK) was applied to both eyes, and mice were placed in dorsal recumbence. Mice remained anesthetized using 2–2.5% isoflurane until the end of the procedure. The jugular area was cleaned three times at patches of skin free of hair, alternating the use of iodine and 70% isopropanol. A ~1 cm incision was made in the skin just off the midline. The underlying tissue was bluntly dissected to isolate the jugular vein to collect ~300 μL of blood using a 1 mL syringe with a 26-gauge needle (CAT# BD 309597, BD Biosciences, Franklin Lakes, NJ, USA). To avoid clotting, syringes had been previously coated with 250 USP units/mL of heparin solution (CAT# NDC 63739-953-25, Eugia Pharma Specialties Limited, Telangana, India). Subsequently, whole blood was centrifuged at 6000 rpm for 2 min at room temperature, and the plasma was collected and immediately stored at −80 °C until further analysis. Plasma volume was replaced with ultra-pure 5% mouse albumin (CAT# IMSALB, Innovative Research, Houston, TX, USA) in sterile pharmaceutical-grade 0.9% NaCl, gently mixed, and reinfused back into the jugular veins of the mice at ~100 µL/min rate (Figure 1). A non-absorbable 6–0 sterile suture was used to close the skin. Animals recovered in a clean cage, providing additional heat sources until fully mobile. The suture was removed 7–14 days post-procedure. Subcutaneous buprenorphine (0.05 mg/kg) and carprofen (5 mg/kg) were provided twice a day for 3 days for pain management. Control animals were exposed to the same surgical procedure and drugs, without replacing the plasma. PE treatment started at 3 months of age and was performed 4 times (one per month). All animals were humanely euthanized at 7 months of age.

### 4.3. Immunohistochemistry

After euthanasia, mice were perfused with a solution of 5 mM of EDTA in phosphate-buffered saline (PBS) at pH 7.4 at 4 °C. Then, their brains were removed and processed as previously described [56], with slight modifications. Briefly, brains were fixed in 10% neutral-buffered formalin solution (CAT# 032-059, Fisher Scientific Company LLC, Houston, TX, USA) for 72 h. Samples were processed in ethanol gradients and paraffin-embedded. Paraffin blocks were sectioned into 10 μm thick sections and processed for immunohistochemistry. Amyloid plaque staining was performed using the anti-Aβ 4G8 antibody (CAT# 800701, BioLegend, San Diego, CA, USA) at 1:5000 dilution. All sections were pretreated with 85% formic acid. Reactivity was detected after 1 h of incubation with a goat anti-mouse HRP-linked secondary antibody and visualized using a DAB Kit (CAT# SK-4100, Vector Laboratories, Newark, CA, USA) as instructed by the manufacturer. Finally, sections were counter-stained with Mayer’s Hematoxylin (CAT# MHS16, Sigma-Aldrich, Burlington, MA, USA). Similarly, the glial fibrillary acidic protein (GFAP) content was analyzed using an anti-GFAP antibody (CAT# Ab7260, Abcam, Waltham, MA, USA) at 1:500 dilution and using citrate buffer for antigen retrieval.

### 4.4. Thioflavin-S Staining

Thioflavin-S (ThS) staining was performed to identify amyloid plaques as previously described [57], with slight modifications. Briefly, deparaffinized/rehydrated samples were incubated with ThS solution (0.025% in 50% ethanol) for 10 min (200 μL/section). The samples were washed two times in 50% ethanol and three times in distilled water for 5 min on a shaking platform. Finally, samples were mounted with Fluorsave (CAT# 345789, Millipore, Burlington, VT, USA), an aqueous mounting media.

### 4.5. Immunofluorescence

Ionized calcium-binding adaptor molecule 1 (Iba1) was used to assess microglial cells. Here, 10 µm sections from formalin-fixed, paraffin-embedded (FFPE) mouse brain tissues were used for staining. Sections were deparaffinized and hydrated in a series of xylenes and ethanol, followed by being washed with PBS. Antigen retrieval was achieved via incubation in 1× citrate buffer (pH 6.0) (CAT# Ab64236, Abcam, Waltham, MA, USA), at 80 °C for 20 min, followed by being washed with PBS. Non-specific binding was blocked using 3% BSA/PBS containing 0.2% TritonX-100 for 1 h. Sections were then incubated overnight in a humidified chamber with the primary antibody anti-Iba1 (CAT# 019-19741, Wako, Mountain View, CA, USA) at 1:500 dilution and prepared in blocking buffer at room temperature for 24 h. Sections were washed in PBS and incubated with Alexa Fluor 594 donkey anti-rabbit secondary antibody (CAT# A21207, Invitrogen, Carlsbad, CA, USA) at 1:500 dilution in PBS-0.2% Triton X-100 for 1 h at room temperature. Then, 300 nM DAPI dihydrochloride staining (CAT# D9542, Sigma-Aldrich, Burlington, MA, USA) was performed to identify nuclei, and samples were mounted using FluorSave mounting media (CAT# 345789, Millipore, Burlington, VT, USA).

### 4.6. Image Analysis

Image quantification was performed in 4 sections per animal, with a distance of 100 μm between sections. Burden quantification was performed in sagittal brain sections, using the cortex and hippocampus as the region of interest (ROI). Brain sections were scanned in bright field (4G8) and fluorescence modes at an excitation wavelength of 430 nm for ThS and 594 nm for Iba1 immunofluorescence using Leica Stellaris^®^ (Leica Microsystems, Buffalo Grove, IL, USA); 10× and 20× scans were digitalized using LAS-X software version 4.4.0.24861 (Leica Microsystems, Buffalo Grove, IL, USA). Image analysis was performed using ImageJ 1.54 software (NIH, Bethesda, MD, USA). To quantify the ThS, Iba1, and GFAP burdens, a threshold was set arbitrarily and kept constant throughout the analysis. Total populations of 4G8-positive plaques were counted, and the diameter was measured in the aforementioned ROIs using ~250 to 500 plaques per ROI in four sagittal brain sections per mouse. Analysis was performed while blinded.

### 4.7. Determination of the Levels of Aβ in Plasma and Brain via Enzyme-Linked Immunoassay (ELISA)

Plasma was acquired by centrifuging whole blood at 2000× *g* for 2 min at room temperature; isolated plasma was then stored at −80 °C until further analysis. After euthanasia, brains were recovered, and the left hemisphere was snap-frozen in liquid nitrogen and preserved at −80 °C. The brain tissue was thawed and weighed to prepare a 10% brain homogenate (*w*/*v*) in PBS plus protease inhibitors (1 tablet in 50 mL of PBS (EDTA-free, protease inhibitor cocktail tablets, CAT# 11873580001, Roche Applied Sciences, Penzberg, Germany)). Samples were homogenized (5000 rpm, 15 s, Precellys 24 Lysis and Homogenization, Bertin Technologies, Montigny-le-Bretonneux, France) as previously described [23]. In total, 200 µL of the brain homogenate was centrifuged at 160,000× *g* for 1 h at 4 °C. The PBS-soluble fraction obtained from this step was promptly frozen using liquid nitrogen and stored at −80 °C until further use. The resultant pellet was dissolved in 100 μL of 70% formic acid (FA) solution and sonicated (five times at an amplitude of 90, for 20 s each time; S-4000, Misonix Inc., Farmingdale, NY, USA). Subsequently, the FA-soluble fraction was separated through centrifugation at 160,000× *g* for 1 h at 4 °C. The supernatant was collected in new tubes and then pH-neutralized (1:20 dilution) with 1 M Tris (pH 10.8). The neutralized FA fractions (1:200 final dilution) were immediately frozen in liquid nitrogen and stored at −80 °C until further use. 

To measure the concentrations of Aβ40 and Aβ42 in PBS and FA fractions of plasma and brain, separate ELISA assays were performed using Human Aβ kits (CAT# KHB3481; KHB3441, Invitrogen, Carlsbad, CA, USA), following the manufacturer’s instructions. Plasma samples and PBS brain fractions were 3-fold-diluted, and the FA fraction was further 10-fold-diluted. These dilutions were prepared in the standard diluent buffer provided with the ELISA kits, following the manufacturer’s recommended protocol for the quantification of Aβ40 and Aβ42 levels. All samples were analyzed in duplicates. After the completion of the assay, the plates were read at OD 450 nm using a microplate spectrophotometer reader (SpectraMax Plus 384, Molecular Devices, LLC, San Jose, CA, USA). Total Aβ levels (Aβ40 + Aβ42) were expressed in pg/mL.

### 4.8. Statistics

The data in this study are expressed as means ± standard error of the mean (SEM). The Kolmogorov–Smirnov test was used to assess whether the data followed a Gaussian distribution. Aβ levels in plasma were analyzed via one-way ANOVA followed by Sidak’s test for multiple comparisons. Aβ levels in the brain, plaque counts, and plaque size data were analyzed using Student’s *t*-test. GFAP and Iba1 burdens were analyzed using the Mann–Whitney test. Comparisons of the plaque size relative frequencies were performed following previously published methodologies [58] using a non-linear fit test after logarithmic transformation of the data. Statistical differences were considered significant at *p* < 0.05 level. All analyses were performed using GraphPad Prism 9.0 software (GraphPad Software Inc., San Diego, CA, USA).

## 5. Conclusions

Plasma exchange (PE) offers an alternative for the non-pharmacological treatment of AD that is more effective and practical compared to the whole-blood exchange methodology in reducing amyloid pathology. However, the precise mechanism through which PE achieves this reduction remains unclear. Based on the findings of this research, it can be speculated that PE may enhance the clearance of Aβ from the brain by facilitating its exit through the blood–brain barrier after reducing the levels of Aβ in the blood. Additionally, albumin replacement during PE may improve the peripheral catabolism of Aβ. It is important to consider that there may be other indirect mechanisms involved in Aβ removal besides the ones mentioned herein. It is worth noting that PE is infrequently used in treating the elderly population, and while there are positive aspects of PE, such as those mentioned above, the procedure has been associated with hemodynamic alterations, skin reactions, and muscle spasms, possibly due to the replacement therapy, the establishment of vascular access, and anticoagulation methods for both young and elderly patients [29,59]. The implementation of PE as a putative intervention for AD requires further development and validation.

## Figures and Tables

**Figure 1 ijms-24-17087-f001:**
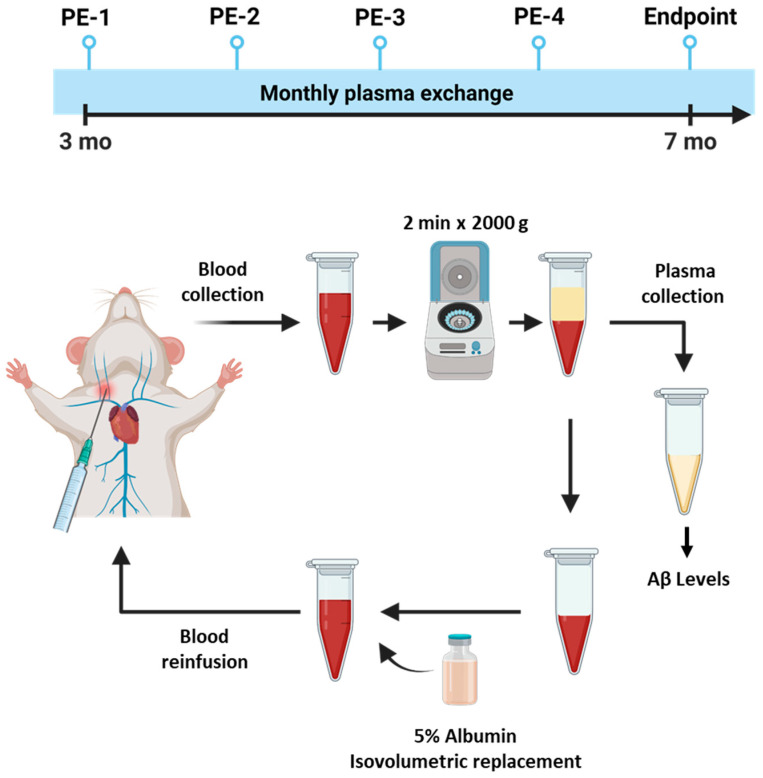
Schematic representation of the monthly PE procedure. Plasma was collected from the surgically exposed jugular vein in fully anesthetized APP/PS1 mice once a month from when the mice were 3 to 6 mo. old (PE-1–4). A total of ~300 μL of whole blood was collected and centrifuged for 2 min at 2000× *g* to further separate and replace the plasma with 5% ultra-pure mouse albumin. The reconstituted blood was reinfused back into the jugular vein. Finally, plasma was also collected at the endpoint at 7 mo. Plasma samples were immediately stored at −80 °C for further analysis.

**Figure 2 ijms-24-17087-f002:**
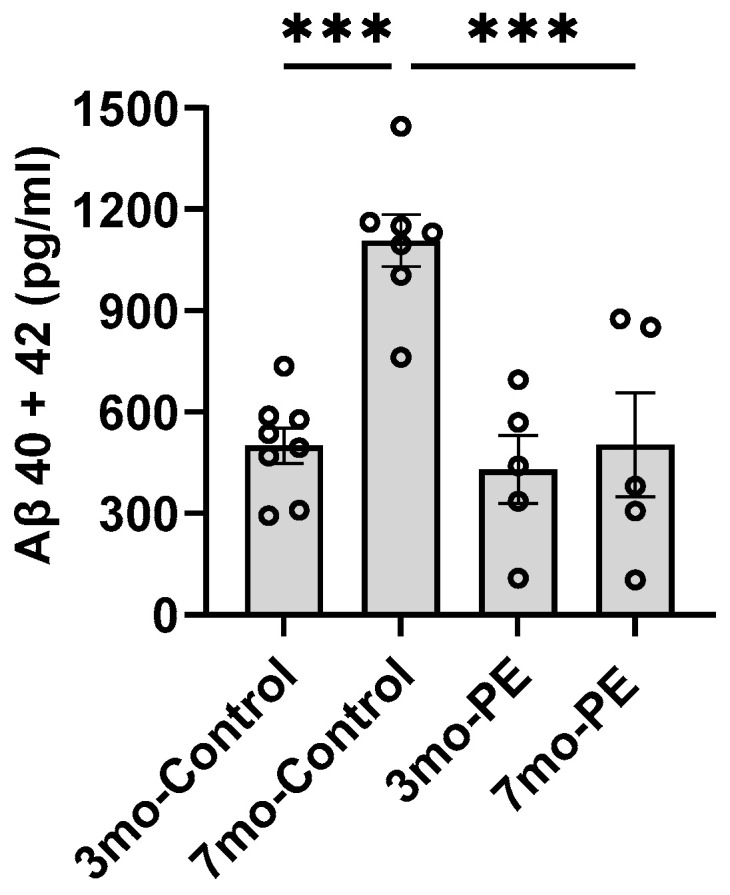
Effect of plasma exchange on the levels of Aβ in plasma. The levels of Aβ in plasma were analyzed in duplicates using an enzyme-linked immunoassay (ELISA). For plasma, baseline and endpoint levels were determined when the mice were 3 months old and 7 months old, respectively. Aβ levels in plasma increased significantly over time: 3 mo-control vs. 7 mo-control (390.6 ± 92.4 vs. 1107 ± 57.92 pg/mL; *p* = 0.0001). PE effectively delayed the progressive increase in Aβ levels in plasma: 7 mo-PE vs. 7 mo-control (498.5 ± 151.8 vs. 1107 ± 57.92 pg/mL; *p* = 0.0006). *n* = 5–8 per group. *** = *p* < 0.001. Graphs represent the mean ± standard error of the mean (SEM).

**Figure 3 ijms-24-17087-f003:**
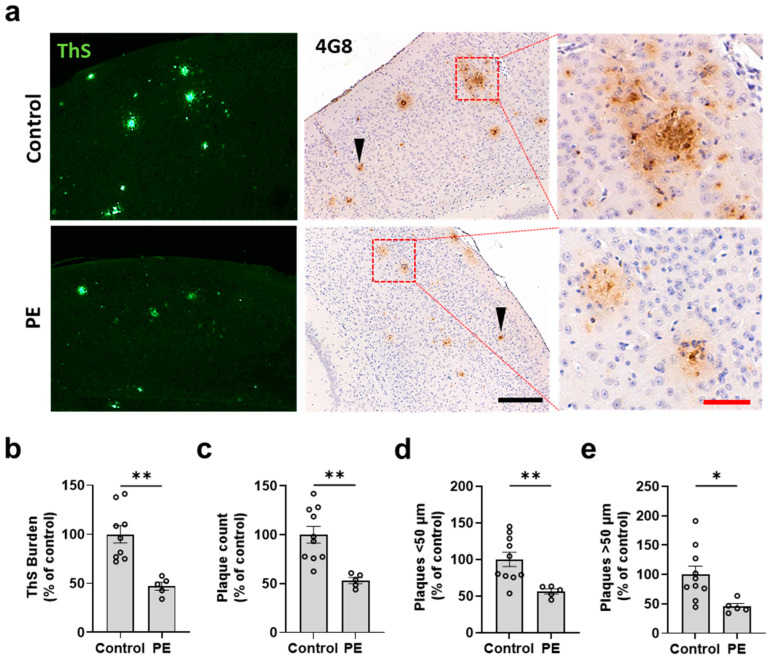
Effect of plasma exchange on amyloid burden and plaque size in the brain cortex. Brain sections were stained with ThS and with 4G8 immunohistochemistry (**a**). ThS amyloid burden (**b**); 4G8-positive plaque count (**c**); and large (>50 μm) (**d**) and small (<50 μm) (**e**) plaques were analyzed and compared between PE and controls. Each symbol represents a different animal. PE reduced ThS amyloid burden (47.18 ± 4.06 vs. 100 ± 8.824%; *p* = 0.0011) (**b**) and 4G8-positive plaque counts (53.22 ± 3.134 vs. 100 ± 8.569%; *p* = 0.0025) (**c**). The plaque size analysis showed that PE treatment was associated with fewer plaques < 50 μm (56.27 ± 3.447 vs. 100 ± 9.829%; *p* = 0.0094) (**d**) and plaques > 50 μm (45.44 ± 5.006 vs. 100 ± 14.19%; *p* = 0.0207) (**e**). The arrowheads show small-sized plaques; red squares show zoomed-in sections. The scale bars are 200 μm (black) and 50 μm (red). *n* = 5–10 per group. Data are expressed as a percentage of control. * = *p* < 0.033, ** *p* < 0.002. Graphs represent the mean ± standard error of the mean (SEM).

**Figure 4 ijms-24-17087-f004:**
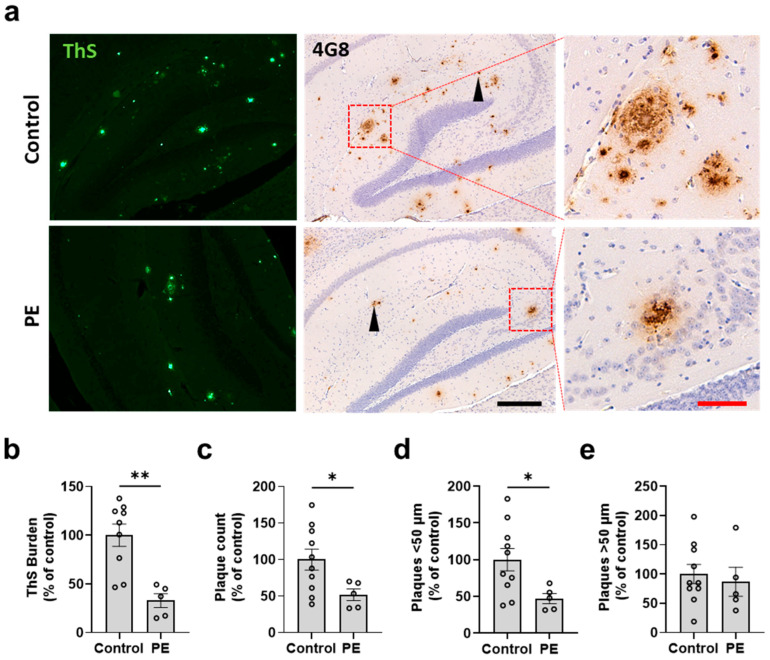
Effect of plasma exchange on Aβ burden and plaque size in the hippocampus. Brain sections were stained with ThS and with 4G8 immunohistochemistry (**a**). ThS amyloid burden (**b**), 4G8-positive plaque count (**c**), and large (>50 μm) (**d**) and small (<50 μm) (**e**) plaques were analyzed and compared between PE and controls. Each symbol represents a different animal. PE-treated mice had reduced ThS burden (32.75 ± 7.071 vs. 100 ± 11.51%; *p* = 0.0016) (**b**); 4G8-positive plaque count (51.84 ± 7.854 vs. 100 ± 14.36%; *p* = 0.0421) (**c**); and plaques < 50 μm (47.16 ± 6.925 vs. 100 ± 15.25%; *p* = 0.0428) (**d**). The arrowheads show small-sized plaques; red squares show zoomed-in sections. No significant differences were found in large-sized plaques. The scale bars are 200 μm (black) and 50 μm (red). *n* = 5–10 mice per group. Data are expressed as a percentage of control. * = *p* < 0.033; ** *p* < 0.002. Graphs represent the mean ± standard error of the mean (SEM).

**Figure 5 ijms-24-17087-f005:**
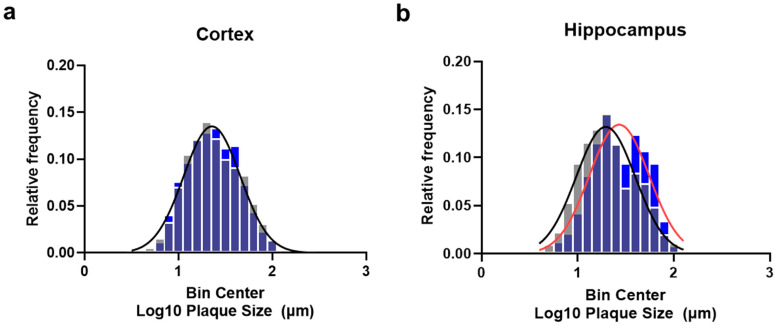
Effect of plasma exchange on the populations of plaques analyzed according to size in the cortex and hippocampus. After the logarithmic transformation of plaque size data from ~250 to 500 plaques per subject (*n* = 5–10 per group), relative frequency distributions were generated at a bin size of 0.1 and analyzed using a nonlinear best-fit model. PE did not modify the relative frequency distribution in the cortex (*p* = 0.2619) (**a**). Meanwhile, in the hippocampus, the frequency distribution was significantly different (*p* = 0.0034) (**b**), showing a reduction in the relative frequency of small-sized plaques and an increase in the relative frequency of large-sized plaques. PE data are represented with blue columns and red lines; controls are represented with gray columns and black lines. Lines represent the best-fit curves.

## Data Availability

The original contributions presented in the study are included in the article/Appendix A; further inquiries can be directed to the corresponding author.

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
