# Peer review of "Plasma Exchange Reduces Aβ Levels in Plasma and Decreases Amyloid Plaques in the Brain in a Mouse Model of Alzheimer’s Disease"

_ijms, 2023, doi:10.3390/ijms242317087_

Round 1

Reviewer 1 Report

Comments and Suggestions for Authors

The authors have presented very interesting data that appear to result from the effect of plasmapheresis in the Alzheimer´s disease mouse model. I appreciate the innovative therapeutic approach. In my assessment, the manuscript is largely composed in a scientifically sound and formally correct manner. However, I have several comments outlined below (the first point is critical and requires clarification):

Major:

  • The manuscript lacks information regarding the number of animals utilized in the experiment, which is a vital component for statistical analyses and interpreting the presented results. The only (contradictory) info is given in the Introduction Page 2, Line 56: “…we subjected male and female APP/PS1 mice” and in the Materials and Methods Page 8, Line 351: “hemizygous male APP/PS1 mice”. Whether the study involved a single plasmapheresis (PE) mouse and a single control mouse or a larger group, this information should be correctly and explicitly provided. The absence of this data necessitates clarification and explanation from the authors.
  • Similarly, raw data for the statistical analyses should be provided.

I have noticed, that the repeatedly referenced previous study (21) that should provide the basis for the current study also lacks information on the number of animals.

Minor:

  • In the Introduction, the statement on Page 1, Lines 40-42 needs reformulation – it is not scientifically correct. Generally, I suggest extending the part about the possible roles of amyloid beta in AD pathology to provide a more relevant background.

·        Plasmapheresis is not without risks, I suggest mentioning and discussing them.

·        Abbreviations like "ThS" and "4G8" should be defined in the text when they are first used, rather than waiting until the last section (Methods) for their explanation.

·        In Figure 1, please specify that it is -80 degrees Celsius.

Author Response

1. The manuscript lacks information regarding the number of animals utilized in the experiment, which is a vital component for statistical analyses and interpreting the presented results. The only (contradictory) info is given in the Introduction Page 2, Line 56: “…we subjected male and female APP/PS1 mice” and in the Materials and Methods Page 8, Line 351: “hemizygous male APP/PS1 mice”. Whether the study involved a single plasmapheresis (PE) mouse and a single control mouse or a larger group, this information should be correctly and explicitly provided. The absence of this data necessitates clarification and explanation from the authors. Similarly, raw data for the statistical analyses should be provided.

We apologize for the ambiguous information and the lack of clarity in the number of subjects per experiment as well as in the number of Plasma exchange procedures. We have now included in each figure legend the number of animals used. We also state that each symbol in the graphs correspond to a different animals, so the number of mice per group can be easily identified by looking at the graphs.

2. In the Introduction, the statement on Page 1, Lines 40-42 needs reformulation – it is not scientifically correct. Generally, I suggest extending the part about the possible roles of amyloid beta in AD pathology to provide a more relevant background.

Thanks to the reviewer for this suggestion. Following the advice we have extended the discussion about possible roles of amyloid beta in AD pathology.

3. Plasmapheresis is not without risks; I suggest mentioning and discussing them:

We agreed and appreciate the reviewer’s suggestion. Literature has been included to describe possible deleterious effects of plasma exchange. Lines 279, and 484 - 488.

4. Abbreviations like "ThS" and "4G8" should be defined in the text when they are first used, rather than waiting until the last section (Methods) for their explanation.

Thanks to the reviewer, we have addressed this in the text as suggested.

5. In Figure 1, please specify that it is -80 degrees Celsius.

Thanks to the reviewer for the detailed observation, we have addressed this accordingly.

Reviewer 2 Report

Comments and Suggestions for Authors

In the manuscript of Ramirez et al, an interesting methodology of plasma exchange to impact Amyloid beta pathology in Alzheimer's disease (AD) is presented in transgenic mouse model of familial AD. Although the manuscript provides a novel approach, it requires improvements in order to be considered for publications. My comments to authors are presented below:

1) In background, please provide current knowledge about the brain clearance of Amyloid beta across the blood-brain barrier and via peripheral organs.

2) It is not clear why the plasma samples were collected only at 3 and 7 months. It would be important to see the time-dependent effect of plasma exchange on Amyloid beta in plasma. Could you provide these data?

3) In Figure 3, total Amyloid beta 40 and 42 concentration in plasma is presented. The titles of the axes are not clear. In addition, please provide the same information for total Amyloid beta 40 and 42 concentrations in plasma separately.

4) It is not clear why 3-month-old animals were used, as at this age the model does not develop main characteristics of Amyloid beta pathology. Which stage of AD does it represent and what would be an approach in clinics taking into account difficulties with diagnosis of AD at early stages?

5) In discussion, please discuss how your findings can be explained in line with the paradigm of brain clearance of amyloid beta across the blood-brain barrier.

Author Response

1. In the background, please provide current knowledge about the brain clearance of Amyloid beta across the blood-brain barrier and via peripheral organs.

We appreciate the comment, we have included recent relevant literature to improve the section about brain clearance of amyloid beta and the role of peripheral organs. Lines 46 to 51.

2. It is not clear why the plasma samples were collected only at 3 and 7 months. It would be important to see the time-dependent effect of plasma exchange on Amyloid beta in plasma. Could you provide this data?

We apologize for the lack of clarity, we have included a precise description of the sampling times (Lines 89 to 94). In supplementary figure 3, the levels of Amyloid beta in plasma are reported for the intermediate time points in the PE-treated animals.

3. In Figure 3, total Amyloid beta 40 and 42 concentration in plasma is presented. The titles of the axes are not clear. In addition, please provide the same information for total Amyloid beta 40 and 42 concentrations in plasma separately.

We have reorganized the information in the graphs and clarified the axes. Now it says (Aβ40+Aβ42) pg/ml. We have also reorganized the order of the bars to make it easier to read and compare. We have expressed our results based on previous literature showing the addition of peptides 1-40 and 1-42.  However, in the attached file are the data as independent graphs for the reviewer perusal.

4. It is not clear why 3-month-old animals were used, as at this age the model does not develop main characteristics of Amyloid beta pathology. Which stage of AD does it represent and what would be an approach in clinics taking into account difficulties with diagnosis of AD at early stages?

The rationale for choosing an early age of the model is to analyze the effect of the treatment at the beginning of plaque pathology. The plan was to evaluate the effect of the treatment in new plaque formation and plaque growth, rather than on elimination of existing plaques. The rationale for age selection is now explained in lines 293 to 298. Moving forward, we will perform experiments to assess the beneficial effect of PE in animals with established pathology.

5. In discussion, please discuss how your findings can be explained in line with the paradigm of brain clearance of amyloid beta across the blood-brain barrier.

We now include a paragraph describing the possibility of Aβ clearance from the brain across the blood-brain barrier and how this relates to our experiments. (Lines 314-318)

Round 2

Reviewer 1 Report

Comments and Suggestions for Authors

Dear authors, thank you revising the manuscript and addressing my comments. However, one comment still remains: I suggest to include the information on the number of animals also in the Materials and Method part (subchapter 4.1).

Otherwise I consider the revised version of the manuscript suitable for publication.

Author Response

Many thanks for the comments. Following the reviewer's advice we have now indicated the number of animals used in the Materials and Methods section

Reviewer 2 Report

Comments and Suggestions for Authors

Ramirez et al have provided all necessary information according to my comments. I would recommend the Editor to accept the manuscript for publication in the current form.

Author Response

Many thanks for your kind comments